# Two Forms of Thick Filament in the Flight Muscle of *Drosophila melanogaster*

**DOI:** 10.3390/ijms252011313

**Published:** 2024-10-21

**Authors:** Hosna Rastegarpouyani, Alimohammad Hojjatian, Kenneth A. Taylor

**Affiliations:** 1Institute of Molecular Biophysics, Florida State University, Tallahassee, FL 32306-4380, USA; hrastegarpouyani@fsu.edu (H.R.); ahojjatian@fsu.edu (A.H.); 2Department of Biological Science, Florida State University, Tallahassee, FL 32306-4295, USA

**Keywords:** aging, flightin, myofilin, paramyosin, stretchin-klp

## Abstract

Invertebrate striated muscle myosin filaments are highly variable in structure. The best characterized myosin filaments are those found in insect indirect flight muscle (IFM) in which the flight-powering muscles are not attached directly to the wings. Four insect orders, Hemiptera, Diptera, Hymenoptera, and Coleoptera, have evolved IFM. IFM thick filaments from the first three orders have highly similar myosin arrangements but differ significantly among their non-myosin proteins. The cryo-electron microscopy of isolated IFM myosin filaments from the Dipteran *Drosophila melanogaster* described here revealed the coexistence of two distinct filament types, one presenting a tubular backbone like in previous work and the other a solid backbone. Inside an annulus of myosin tails, tubular filaments show no noticeable densities; solid filaments show four paired paramyosin densities. Both myosin heads of the tubular filaments are disordered; solid filaments have one completely and one partially immobilized head. Tubular filaments have the protein stretchin-klp on their surface; solid filaments do not. Two proteins, flightin and myofilin, are identifiable in all the IFM filaments previously determined. In *Drosophila*, flightin assumes two conformations, being compact in solid filaments and extended in tubular filaments. Nearly identical solid filaments occur in the large water bug *Lethocerus indicus*, which flies infrequently. The *Drosophila* tubular filaments occur in younger flies, and the solid filaments appear in older flies, which fly less frequently if at all, suggesting that the solid filament form is correlated with infrequent muscle use. We suggest that the solid form is designed to conserve ATP when the muscle is not in active use.

## 1. Introduction

The flight muscles of *Drosophila melanogaster*, the common fruit fly, are a valuable model system for the study of muscle structure and function due to their high structural organization, ease in generating transgenic organisms, and tolerance of drastic changes in their composition and structure [1,2,3]. *Drosophila*’s indirect flight muscle (IFM) is specialized for generating wing beats at frequencies exceeding 200 Hz [1] while exhibiting remarkable structural regularity compared to other insect muscle fibers [4].

The primary component of thick filaments is myosin II. However, several other associated proteins are also involved. Paramyosin and miniparamyosin (~74 kDa), which originate from the same gene [5,6], play significant roles in determining the length and diameter of thick filaments [7]. Paramyosin is an α-helical coiled coil protein found in the thick filaments of all invertebrate muscles and is typically located within the central core region [8]. Paramyosin function remains enigmatic, with hypotheses suggesting a mechanical function in enhancing structural stability during tension development or an influence on the ATPase activity of contractile proteins [7,9,10]. These hypotheses suggest that paramyosin interacts with the myosin molecules within thick filaments, but the precise effects on myosin arrangement within the filaments remain unclear [11].

Several non-myosin proteins play significant roles in both thick filament function and assembly. Flightin, a 20 kDa phosphoprotein, is distributed throughout the thick filament except at the M-line and A/I junction. Flightin is crucial for maintaining structural organization, contributing to stiffness and facilitating stretch-activation in IFM fibers [12,13,14]. Myofilin is another non-myosin protein associated with the thick filament core, but its function is not yet defined [15]. Stretchin-klp, a 225 kDa non-myosin protein translated from an alternatively spliced transcript of the *Drosophila Stretchin-Mlck* gene [16,17], is located on the outside surface of the filament backbone [18,19]. In addition, a pair of large, titin-like molecules, projectin and kettin, connect thick filaments with thin filaments emanating from the Z-band, thereby contributing to the muscle’s passive stiffness [20,21].

Four insect orders have developed IFM: Coleoptera (beetles), Diptera (flies), Hemiptera (true bugs), and Hymenoptera (bees and wasps). All four orders diverged from a common ancestor ~ 370 million years ago (Mya) [22]. Hemipterans diverged first at 340 Mya, followed by diversions leading to Hymenoptera and Coleoptera. Dipterans diverged last at 170 Mya. High-resolution cryoEM studies of species from three of these orders have been reported [18,19,23,24,25] revealing filaments with either systematically ordered or disordered heads, three different arrangements of non-myosin proteins, and variable paramyosin visibility in the filament core. However, the myosin tail arrangements are very similar. Having diverged from a common ancestor, can the different species make IFM myosin filaments like any of the others at any point in their life cycle?

Here, we report the cryoEM imaging of two different IFM thick filament types isolated from the Dipteran *Drosophila melanogaster*. The work is an improvement over previous reports in reaching a resolution of 2.8 Å for both reconstructions and revealing two conformations for the non-myosin protein flightin. The second filament type is completely novel for *Drosophila*. One filament type is characterized by disordered myosin heads, stretchin-klp on the backbone outer surface, and an “empty” filament core like previous *Drosophila* reconstructions [18,19]. These filaments we call “tubular”, not to be confused with the term “tubular muscles” which constitute the bulk of the other striated muscle of *Drosophila* [26]. The other filament type has ordered myosin heads, no visible proteins on the backbone surface, and a full paramyosin core like that seen in the Hemipteran *Lethocerus indicus* [23]. These filaments we call “solid”. In addition to the distinct protein compositions and head conformations between the two filament types, flightin differs in its conformation. Variations in the structure of thick filaments potentially result from differences in the quantities of non-myosin proteins and post-translational modifications, possibly due to aging, which lead to distinct physiological characteristics specific to that muscle. To the extent that these two thick filament structures represent two distinct fiber types, the results suggest that even a specialized muscle such as IFM can change filament type in step with changes in insect physiology.

## 2. Results

### 2.1. Two Forms of Indirect Flight Muscle Myosin Filaments

To increase the resolution of *Drosophila* IFM thick filament reconstructions beyond the previous best of 4.7 Å [19], we prepared a cryoEM sample under relaxing conditions (see Materials and Methods) and collected a dataset consisting of 16,500 movies. The filaments had disordered heads and a tubular (hollow) backbone consistent with previous observations [18,19], but the highest resolution achieved was only 3.8 Å.

A new sample was prepared to increase the segment count and potential resolution. Because of a time constraint imposed by an impending data collection window at NCCAT, muscle for this specimen was obtained from flies without considering age or other conditions, except to exclude the generally smaller male flies.

Surprisingly, the initial screenings of micrographs from this specimen revealed an ordered arrangement of myosin heads on filaments with solid backbones (Figure 1A), like the distinctive flight muscle thick filaments of *Lethocerus indicus* [23] but unlike the tubular backbones typically seen in *Drosophila*. We will refer to these as “solid filaments”. We collected and reconstructed a dataset of 39,660 movies of these solid filaments.

The solid filament dataset was not completely homogeneous. Some images showed the presence of filaments with both ordered heads and disordered heads (Figure 1B). Because it takes thoraces from 10–15 flies to make a thick filament preparation, we do not know if the solid and tubular filament types represent inter- or intra-fly variation.

Alerted to the presence of variability, we collected a third dataset of 27,365 movies, but this time from 30-day-old “middle-aged” flies, to investigate the potential impact of aging on filament conformation. The initial screening of these filaments (Figure 1C) revealed that most filaments had disordered heads and tubular filament backbones. Like the solid filament images, a portion of the tubular filament images appeared intermediate between ordered or disordered.

The helical angles of the tubular and solid filaments were similar, at 33.88° and 33.97°, respectively. Additionally, the helical rise for each was nearly identical, measuring 150 Å for the tubular and 149.33 Å for the solid filaments. The pixel size was scaled to make the helical rise 145 Å for both, as measured by X-ray fiber diffraction [27].

### 2.2. 3D Reconstruction and 3D Classification

The classification of projections from the two samples clearly displays differences (Figure 1D–F). Density extending outward from the solid filaments is strong and perpendicular to the filament backbone (Figure 1D). Additionally, a faint signal for the tilted proximal S2, defined as the initial segment of the myosin α-helical coiled coil, of the solid filament is visible in the 2D classes (Figure 1D, arrow).

Classification of the solid filaments resulted in two dominant classes. Two examples show ordered heads and a solid backbone (Figure 1E, top), while two other examples totaling 11,402 segments show no ordered heads and tubular backbones (Figure 1E, bottom). The subsequent 3D refinement of the tubular filaments yielded a resolution of 7.88 Å (see below). Classification of the tubular filaments shows a weak head signal (Figure 1F).

The highest global resolutions achieved for the consensus maps (from the dominant class averages) were 2.8 Å for the solid filaments (Appendix A) and 2.8 Å for the tubular filaments (30-day-old flies) (Appendix A). The following discussion focuses on the 2.8 Å reconstructions, which are sufficient for the description of the differences because they have equivalent resolution.

### 2.3. Multiple Differences Between Filament Types

The 3D reconstructions confirmed the impressions from the 2D projections regarding the ordering of the heads and the solidity of the backbone (Figure 2A,B). Segmentation of both tubular and solid filament reconstructions identified the non-myosin proteins flightin, myofilin, stretchin-klp, and paramyosin (Figure 2A–C). Comparing both maps broadly, the non-myosin densities among the myosin coiled coil tails have both similarities and differences.

A noticeable difference between the two reconstructions is a density along the backbone surface of the tubular filaments, which is from the non-myosin protein stretchin-klp (Figure 2A). Stretchin-klp density is missing from the backbone of the solid filaments (Figure 2B). Conversely, paramyosin density is observed in the core of the solid filaments (Figure 2B), while the tubular filament cores of the disordered-head filaments appear empty (Figure 2A). In the tubular filaments identified in the solid filament dataset, this three-density pattern is visible, usually as a pair of Ig-like domains in the same location as those in the high-resolution reconstruction (Figure 2C).

Paramyosin and stretchin-klp appear inversely correlated. When stretchin-klp amounts are high and the heads are predominately disordered, paramyosin is present in a low quantity (Figure 2A). When stretchin-klp levels are low, paramyosin levels are high, and the myosin heads are ordered (Figure 2B). Differences are also found in the non-myosin proteins flightin and myofilin.

### 2.4. Two Conformations for Myofilin and Flightin

In tubular filaments, the myofilin density is smaller than predicted by the amino acid sequence (Figure 2D) and comparable in size to that observed in previous tubular *Drosophila* thick filament reconstructions [18,19]. Conversely, in the solid filaments (Figure 2E). the myofilin density is comparable in size and shape to that of the solid thick filaments of *Lethocerus* IFM [23]. The significance of the size difference is not known because myofilin has not been studied genetically or biochemically at the same level of detail as flightin.

The two conformations of flightin encompass changes in several domains. Previous work with *Drosophila* flight muscle showed a connection between the WYR domain and the flightin C-terminal domain in tubular filaments [19]. That connection is incomplete in the tubular filaments shown here. Instead, extended but incomplete densities are observed at the end of the WYR domain and the beginning of the C-terminal domain, but they do not connect (Figure 3A). However, these incomplete densities fall within the density envelope of the previously reported flightin structure [19].

In the solid filaments, extra density is seen near the WYR domain and the beginning of the C-terminal domain but not in the same location as that found for the tubular filaments. These extra densities poorly overlap with the flightin densities seen in the tubular filaments (Figure 3B). At the WYR domain, a different arrangement of mass suggests a direct connection between the WYR domain and the nearest C-terminal domain.

We interpret the lack of a clear flightin difference between solid and tubular filaments as due to flightin heterogeneity in the two averaged structures. That is to say, the present solid filament reconstruction with ordered heads has some contamination from segments with poorly ordered heads, possibly affecting flightin homogeneity and vice versa. However, the small differences suggest that flightin can assume two conformations, one of which is seen in tubular filaments (disordered heads), in which the WYR and C-terminal domains are linked by a mostly extended connector (Figure 3C) as previously observed [19].

In the solid filaments (ordered heads), e.g., *Lethocerus*, no evidence is seen of an extended connection between the WYR domain and the C-terminal domain seen previously [19]. The location of the flightin C-terminal domain is the same so far in all of the IFM thick filament structures. It is the connector that is changing. Evidence for this comes from the densities about the WYR domain in solid filaments that are not seen in tubular filaments (yellow arrowhead in Figure 3B). These densities suggest a different connection, one that links the WYR domain to its nearest C-terminal domain. We refer to the flightin conformation in tubular filaments as ”extended flightin” (Figure 3C) and in solid filaments as “compact flightin” (Figure 3D). A low-pass-filtered map of the compact flightin shows it literally wrapping around a single myosin coiled coil (Figure 3E). How much difference there is in the strength of the flightin–myosin interaction is uncertain because the two contributing domains, WYR and C-terminal, are in the same location. It is only their connection that differs.

The external domain of *Drosophila* flightin encompasses residues A2-G74. In tubular filaments, only residues starting from P69 and after are ordered mostly due to embedding within the myosin tail annulus [19]. In the solid *Drosophila* filaments, the proximal S2 is ordered and swung azimuthally across the backbone surface due to binding of the free head to the filament backbone surface [23], and the flightin external domain is visible up to residue P66, stabilized by a proximal S2 contact.

### 2.5. Myosin Head and Proximal S2

The relaxed *Lethocerus* thick filament is characterized by well-resolved myosin heads in an interacting heads motif (IHM) arrangement [23]. That IHM orientation is also observed in *Drosophila’s* solid filaments, mirroring the exact orientation seen in the *Lethocerus* structure (Figure 2B and Figure 4A–C). The unusual IHM orientation seen in insect IFM has the “free” head bound to the filament backbone with the “blocked” head oriented as if to “pin” the free head in its place. Neither head is particularly well ordered, but the blocked head appears the more dynamic of the two and is generally not visible without severe low-pass filtering.

As anticipated, the tubular filaments show very little head density, likely because of head mobility. In the prior, tubular *Drosophila* filament studies [18,19], four isolated densities appear at the usual axial locations of myosin heads but remain unconnected or barely connected to the proximal S2 (Figure 4E,G). At similar resolution cutoffs applied to the solid filaments, the free head is visible, but the blocked head is not (Figure 4A,B,D).

A superimposition of the solid and tubular filaments shows an alignment of two densities attributable to myosin heads—one with a distinct shape and the other shapeless (Figure 4A). When the maps are aligned based on their very similar backbone densities, the proximal S2s of both structures clearly emerge from the same axial and azimuthal positions on the backbone. The proximal S2 in the tubular filament follows a nearly axial path (Figure 4A), consistent with findings from previous studies [18,19]. In contrast, the proximal S2 in the solid filaments is bent azimuthally by 17° (Figure 4A), as observed in *Lethocerus* [23]. The proximity of the “floating” density to the beginning of the proximal S2 suggests that it may represent the average density of disordered myosin regulatory light chains.

### 2.6. Stretchin-klp Affects Myosin Head Ordering

Stretchin-klp, a ~215 kDa protein (purple in Figure 4E,F), was first identified in *Drosophila* flight muscle A-bands [17] and structurally in a 7 Å thick filament reconstruction [18]. Stretchin-klp binds along the filament shaft but avoids bare zones or filament tips. The amino acid sequence consists of five repeats of the pattern Ig-like—short linker – Ig-like—long linker [18,19]. The Ig-like domains and short linkers are similar, whereas the long linkers are highly variable and occasionally large enough to form small, folded domains. Stretchin-klp follows a left-handed helical track along the filament’s backbone surface (Figure 4F), a feature absent in *Lethocerus indicus* and *Bombus ignitus* [19,23].

The stretchin-klp helical tracks pass under the location where the IHM “free” head binds the filament backbone, consistent with the proposed role for stretchin-klp in preventing the IHM “free” head from binding to the filament backbone, thereby stabilizing its conformation (Figure 4). This hypothesis is supported here by the absence of these densities in solid filaments. Thus, stretchin-klp’s presence correlates with disordered myosin heads, highlighting its significant role in filament organization.

### 2.7. Paramyosin Visibility Is Highest in Solid Filaments

Solid filaments obtain their appearance from paramyosin filling much of the filament core inside the myosin tail annulus. Tubular filaments lack this feature so the volume inside the myosin tail annulus generally appears empty. Tubular filaments with disordered myosin heads and stretchin-klp have a 180 Å diameter empty core (Figure 2A). In contrast, the solid filaments feature eight densities attributable to paramyosin and a 50 Å diameter empty core (Figure 5). Thick filaments with solid cores of paramyosin are observed across different species including *Lethocerus* and *Bombus* [23,25]. However, the visibility of paramyosin does not correlate directly with the ordering of myosin heads; *Lethocerus* has ordered heads and *Bombus* does not, though both show a paramyosin core.

Here, the paramyosin core shows more detail than previous observations. In some places, the left-handed pitch of the α-helical coiled coil is visible (Figure 5B). In addition, distinct near-horizontal stripes at ~4.5 Å resolution, which come from the α-helix pitch, are also visible (Figure 5A,B), consistent with the higher resolution in the current study.

Paramyosin molecules have a length of 1220 Å or 8.4 crowns (1 crown = 145 Å) [28]. In the paramyosin P1 lattice, adjacent paramyosin molecules are offset 725 Å or 5 crowns. Because the averaging in the reconstruction occurs over axial lengths of a single crown, paramyosin will not be accurately represented in the reconstruction. However, some features may be informative. The excess length of 0.4 crowns combined with the helical averaging over a single crown means that a 60 Å segment of the paramyosin density should have greater density than the remaining 85 Å. In previous reconstructions, the region of added density was located at the level of the myosin heads [19,23]. The same pattern is also seen here (Figure 5A) (see also a more detailed description in the Supplementary Information of [23]).

Potential interactions between flightin, myofilin, and paramyosin are observed. Paramyosin has been shown to be important in determining filament length in *Caenorhabditis elegans* [29]. Its role in determining filament length in *Drosophila* is less clear. Flightin, the only *Drosophila* protein shown to affect thick filament length [13], makes a close approach to paramyosin through the connecting loop between the α-helices of the WYR domain (Figure 5D,E). The flightin C-terminal domain, though nearby, does not contact paramyosin. A region of myofilin also comes close to the paramyosin densities but not as close as the WYR domain of flightin (Figure 5F,G). Some interaction between myofilin and paramyosin is implied by the greater amount of myofilin that is seen in those species whose filaments have visible paramyosin density.

## 3. Discussion

This study was undertaken to obtain an atomic resolution structure of a thick filament from an insect species utilizing indirect flight muscle. Invertebrate thick filaments are ideal for this study because the filaments generally have a helical structure [30], which vertebrate filaments lack [31,32,33]. Subnanometer resolution thick filament structures have been obtained from species of three insect orders, all of which had very similar myosin tail arrangements coincident with significant differences in non-myosin proteins [18,23,25]. The most useful atomic model would be from the Dipteran *Drosophila melanogaster*, which is a genetic model organism with particular utility for the study of striated muscle. Two previous cryoEM studies of *Drosophila* thick filaments had disordered myosin heads [18,19]. The present study extended the resolution beyond 3 Å and identified a different *Drosophila* thick filament form, one with ordered myosin heads.

Earlier studies revealed two distinct IFM myosin filament types, either tubular or solid, sometimes in combination within the same sarcomere [34]. Transverse IFM electron micrographs from the flesh fly *Phormia terraenovae* displayed both solid and tubular filaments, while the waterbug *Lethocerous uhleri* and honeybee *Apis mellifica* exclusively exhibited solid filaments [11]. Solid filaments have paramyosin contents up to 11%, while tubular filaments typically contain around 2% paramyosin [35].

### 3.1. Changes in Four Non-Myosin Proteins Correlate with Filament Structure

Differences in four non-myosin proteins correlate with changes in myosin filament structure. The two most obvious changes occur with paramyosin and stretchin-klp, whose visibility is anticorrelated. Paramyosin is present in such a low quantity in tubular filaments it is not visible in the symmetry-imposed reconstruction. Solid filaments show a complete annulus of paramyosin in the filament core. Conversely, stretchin-klp, which is easily “visible” when paramyosin is “invisible”, is “invisible” when paramyosin is “visible”.

Flightin was initially discovered in *Drosophila* [12] and visualized structurally in *Lethocerus* thick filaments [23]. The flightin densities observed in the three species so far studied (*Lethocerus*, *Bombus*, and *Drosophila*) share a similar shape. An external domain is present that is mostly disordered, but when the myosin heads are ordered, it is longer but still incomplete. The external domain is followed by a V-shaped structure consisting of an extended polypeptide chain followed by a prominent folded domain in the middle, the WYR domain, with a helix-loop-helix structure [19]. The WYR domain is followed by an extended connector and a small, folded domain at the C-terminus, previously dubbed the “blue” protein because of the color used to display its shape [23]. The V-shaped structure, WYR domain, and C-terminal domain have so far been consistently visualized in the same location in all the IFM thick filament reconstructions reported so far. The connector between WYR and C-terminal domains has so far only been seen fully resolved in the latest *Drosophila* tubular reconstruction [19] and in the *Bombus* solid filaments, both of which had disordered myosin heads [25]. In *Lethocerus*, the connector is also absent, although the myosin heads are ordered [23]. Is there an underlying pattern to these differences?

Flightin changes structure subtly. It is present in quantity in both solid and tubular filament forms; it differs in structure in both forms. We know from previous work on both *Drosophila* and *Bombus* that when the myosin heads are disordered, the WYR and C-terminal domains are linked by an extended connector domain [19,25]. Thus, the flightin extended conformation correlates with disordering of the myosin heads but not with paramyosin visibility in the core, which forms a complete annulus in *Bombus*.

In solid filaments, flightin appears to assume a compact conformation that links the WYR domain with the closest C-terminal domain. The link is not clearly resolved here, but the structure of the C-terminal and WYR domains is distinctly different in solid filaments. A credible atomic model for the short link between WYR and C-terminal domains can be built.

Myofilin, a protein whose role in thick filaments has yet to be defined, differs in visibility between solid and tubular filaments. More myofilin length is visible in solid filaments where more paramyosin is present in the filament core than when no paramyosin is visible, though some paramyosin must be present.

### 3.2. Changes in Filament Structure May Correlate with Age

Insect flight muscle is energetically demanding, constituting a substantial portion of body mass and a considerable investment in terms of protein content [36]. As flies age, various changes become evident in their flight muscle performance [37]. We suggest that the thick filament structures described here, despite the uncertainty in the age distribution of the samples, are most easily interpreted as part of age-related alterations in the flight muscle. Whether they are part of changes programmed into the fly life cycle or are a response to other changes is unclear.

Specific proteins unique to IFM, such as flightin, undergo age-related phosphorylation changes [38]. Miniparamyosin, a protein whose structure and function are poorly defined, is specific to pupal/adult muscles [5]. Its overexpression in *Drosophila* flight muscle produces age-dependent effects such as the development of a flight-impaired phenotype [39].

The work on aging effects in *Drosophila* IFM from the early 1970s found primarily changes in the structure of mitochondria, which were quite pronounced [40,41]. However, another study utilizing the common house fly *Musca domestica* found no changes in either mitochondria or myofibril structure in male flies up to 19 days of age [42].

EM studies conducted on other Dipterans, e.g., *Musca* or *Calliphora*, found dramatic changes in the flight muscle mitochondria but did not note any changes in the myosin filaments [43]. Most transverse section micrographs in that report showed tubular profiles up to 34 days old, but some appeared to show mixtures of solid and tubular filaments, e.g., Figure 4B,C from 34-day-old flies. A later study using the fleshfly *Phormia terrae-novae* found mixtures of solid and tubular thick filaments in the same sarcomere [35], but the age of the fly sample, if known, was not specified.

The present work suggests that over time, adult *Drosophila* thick filaments convert from being exclusively tubular to mostly solid. Our first reconstructions of *Drosophila* thick filaments were predominately from flies 7 days after eclosion [18,19]. In the present work, we believe the solid thick filament structure comes from flies substantially older than the 30-day age of the high-resolution tubular thick filaments. Both high-resolution solid and tubular filament datasets had sufficient filaments of the other type to produce a subnanometer resolution reconstruction, meaning the filament preparations were heterogeneous.

An early study of flight muscle development in pupating *Calliphoria* showed predominately solid thick filaments, which converted to tubular thick filaments by adulthood [44]. A later study of *Drosophila* flight muscle development found that early in pupation, developing IFM myofibrils begin as short sarcomeres of solid filaments. By eclosion, these fibers had converted to be exclusively tubular with a normal sarcomere length [45]. Whether these short, solid filaments are just shorter versions of the long, solid filaments observed here is unknown. The filaments in both studies remained the same diameter, whether solid or tubular, indicating that mass is lost from the solid filaments without a change in filament diameter.

That *Drosophila* thick filaments might change structure with age was suggested by one study that compared young and aged flies and quantified several effects pertinent to the present work [37]. X-ray fiber diffraction showed that the myosin heads moved toward the thin filaments (to a higher radius) with age. The flight muscle fibers became stiffer with age but also showed increased performance. As the flies aged, their mitochondria became “degraded”, suggesting that ATP production was reduced. We think those results are consistent with the present observations.

The unusual orientation of the IHM, first seen in *Lethocerus* and now seen in *Drosophila* flight muscle, positions the blocked-head motor domain at a distance from the thick filament backbone with the free head stabilized on the thick filament backbone [46]. The blocked-head motor domain radius in *Lethocerus* is such that it can contact the actin filament even without dissociating from the backbone-bound free head [46]. This might produce additional stiffness. Although not modeled here, thick filaments with disordered heads would likely position both heads at an intermediate radius between thick and thin filaments, basically the average over all possible orientations of the heads when anchored to the thick filament backbone via the proximal S2 and with variable contacts with the thin filament. The small amount of density at the end of the proximal S2 in the present work possibly represents the regulatory light chain, suggesting most of the head density, although more massive, is spread over a much larger volume. When the heads are ordered, the free head acts as a stabilization site for the blocked head, whose orientation perpendicular to the filament axis stabilized by the free head might more effectively resist compression.

Miller et al. [37] also observed that flight muscle fibers become stiffer with age, but they attributed that to changes in the two proteins that make up the connecting filaments, projectin and kettin. Our work suggests that the thick filament structure also changes with age. The completion of the paramyosin annulus is only indirectly involved, but it is coupled to the ordering of the heads. The IHM blocked head becomes ordered in a way that facilitates contact with the thin filament without detachment from the IHM. What we do not know for certain is how our observation correlates with age.

### 3.3. Four Thick Filament Proteins Could Be Agents for Change

Four proteins change in step with thick filament structure: flightin, myofilin, paramyosin, and stretchin-klp. Is any one of them the trigger for change or do they all change together?

The reconstructions, helically averaged over the crown repeat with 4-fold symmetry enforced, do not display paramyosin and stretchin-klp accurately. Paramyosin in the so-called P1 lattice has a repeat length of five crowns, not one crown [28]. Averaging over single crowns will not represent paramyosin accurately. When paramyosin levels are low, such as in the tubular filaments [35], its visibility is unlikely in an average reconstruction.

We observed stretchin-klp only on the surface of tubular *Drosophila* filaments (Figure 4E,F). The thick filament surfaces of *Lethocerus* [23] and solid *Drosophila* filaments, both with ordered heads, are free of surface proteins (Figure 2B). Conversely, the thick filaments from *Bombus* [25] have disordered heads but have nothing on the surface that can obviously cause the disordering. Thus, there must be more than one method for inhibiting IHM formation on the surface of IFM myosin filaments.

Stretchin-klp has an N-terminal domain of 455 residues followed by five pseudo repeats of Ig-like—short linker – Ig-like—long linker. In previous filament preparations, the N-terminal domain was shown lost after calpain digestion but was present in the myofibril preparation [18]. The remaining protein has a five-crown repeat like that of the paramyosin P1 lattice [19]. Averaging over a single crown averages heterogeneous domain conformations, especially for the long linker, thereby blurring subtle conformational changes in stretchin-klp as a driver for the change in filament structure.

Myofilin’s role in thick filament structure is unknown, but it might be involved in filament length determination. More myofilin density is seen in solid filaments than in tubular filaments, but this increase still leaves much of the myofilin unresolved. What is seen seems not to involve stabilization by paramyosin contacts. Myofilin visibility might simply be due to the completeness of the paramyosin annulus, which reduces the myofilin diffusion volume. Confinement to a smaller volume may promote a relatively low probability folded form.

Numerous studies have shown that flightin is necessary for normal IFM thick filament structure and function [47]. Protein phosphorylation often plays an activating role in many cellular processes [48]. *Drosophila* flightin has five well-characterized phosphorylation sites located at residues S139, S141, S145, T158, and S162 [49]. The two most essential, T158, S162, are located on the connector domain, residues S145-Y166 [19] linking the WYR and C-terminal domains. In the pupal stage of development, flightin is expressed but is not phosphorylated. Flightin phosphorylation increases dramatically following eclosion [38].

The case for flightin phosphorylation as a driver of thick filament form is not compelling. First, the electron microscopy indicates that at eclosion, the flight muscle filaments are tubular [44,45], but the biochemistry indicates that flightin phosphorylation levels are low [38]. Thus, tubular filaments can be constructed without flightin phosphorylation. Second, flightin occupies a crowded position within the myosin tail annulus, and consequently, phosphorylation or dephosphorylation within filaments may be difficult due to obstruction. Tubular filaments, which have a 180 Å empty core, might permit phosphorylation/dephosphorylation of the connector, but the increased paramyosin in solid filaments, with a 50 Å diameter core, combined with little space between the WYR and C-terminal domains, would not. In this context, it is worth noting that the *Lethocerus* ortholog of flightin was originally discovered as a Z-disk protein, hence the name Zeelin 2, and later found to migrate to the Z-disk in glycerinated fibers [50], suggesting a soluble pool that might facilitate continuous exchange. The clear images of well-segregated tubular and solid filaments in the myofibrils of *Phormia terrae-novae* [35] would suggest de novo filament formation rather than a possible random conversion of existing filaments.

### 3.4. What Advantage Does the Change in Filament Form Provide?

Our previous studies on *Drosophila’s* IFM thick filaments at somewhat lower resolutions consistently revealed a disordered arrangement of myosin heads [18,19]. Despite the high similarity in myosin sequences between *Lethocerus* and *Drosophila*, with an 88% identity and 98% similarity, there has been a clear difference: *Lethocerus* shows an ordered arrangement of myosin heads, while *Drosophila* shows disordered heads. The IHM is a common feature in myosin II observed across various cells, tissues, and species, indicating its universal occurrence in most multicellular organisms during their life cycles [51].

Formation of the IHM, at least in vertebrates, is associated with a state of relaxation called the super-relaxed state, which results in a 10-fold reduction in the basal ATPase rate (not the actin-activated rate) in those muscles, for which the effect has been investigated [52,53]. Thus, super relaxation and ordered myosin heads in animals are likely associated with a need to conserve ATP. Super relaxation could play an important role in insect IFM. As suggested previously [25] for the ordered heads in the flight muscle filaments of the large water bug *Lethocerus indicus*, a potential super-relaxed state makes sense for a species that spends most of its time under water where its wings and flight muscles are useless. For flies, if the ordered-head filament form correlates with age, it might make sense to consume less ATP as mitochondrial ATP output degrades [37,40,41,43]. The bumble bees studied in earlier work [25] were likely foragers captured in the wild. In honeybees, the old bees of the hive are usually the foragers [54], which move constantly from one pollen/nectar source to another. An ordered-head thick filament form may not be particularly useful because when the bees stop to gather nectar or pollen, they must still maintain their thoracic temperature above 40 °C to beat their wings at the necessary frequency [55], which they can do by cleaving ATP without producing work. They can also produce heat using a process known as shivering. An ordered-head arrangement in relaxed thick filaments does not prevent flight; large water bugs can fly when they need to but require an additional step, warm up, before flight can occur [55].

## 4. Materials and Methods

### 4.1. Specimen Preparation

Flight muscle was obtained from the thoraces of approximately 10 to 15 *w^1118^* wild-type female flies. Male flies, which are 30% smaller, were always excluded. All flies were maintained at 25 °C with a 12-hour light/12-hour dark cycle and fed a standard cornmeal-based diet. The diet was prepared using 33 L water, 237 g agar, 825 g dried deactivated yeast, 1560 g cornmeal, 3300 g dextrose, 52.5 g Tegosept dissolved in 270 mL 95% ethanol, and 60 mL propionic acid [56]. The thoraces were dissected under a stereomicroscope, following the removal of wings and legs. After dissection, the thoraces were immediately placed into a relaxing buffer (80 mM KCl, 20 mM MOPS, 5 mM MgAcetate, 5 mM ATP, 5 mM EGTA, 1 mM DTT, pH 6.8) to preserve muscle integrity. The muscle was subsequently homogenized by pipetting up and down several times using a P100 pipette.

Myofibrils were separated from solubilized proteins by centrifugation at 6000 rpm for 2 min. The resulting pellet was resuspended in the relaxing buffer containing 1% Triton and then incubated on ice for 15–30 min. Subsequently, the myofibril suspension was centrifuged at 6000 rpm for 2 min, followed by resuspension in the relaxing buffer and pelleted to remove the Triton. The resulting pellet was resuspended in 0.1 mL Calpain buffer (relaxing buffer plus 5 mM CaCl_2_, pH 6.8) to which 1 μl of Porcine Erythrocyte calpain-1 (Athens Research & Technology, Inc. Athens, GA, USA) at a concentration of 1.9 mg/mL was added. Digestion was carried out at room temperature for 1 hour and halted by adding 0.2 mL of Stop buffer (relaxing buffer plus 10 mM EGTA, pH 6.8).

The digested myofibrils were separated by centrifugation at 7000 rpm for 2 min, and the supernatant was discarded. Depending on the pellet size, 35–80 μl of Shear buffer was added to the pelleted myofibrils. The myofibrils were sheared 10–12 times by passing the preparation through a 1 mL syringe with a 26G needle. Large residual solid material was removed by centrifugation at 3500× *g*. The resulting supernatant was collected and mixed with 2 mg/mL of calcium-insensitive gelsolin [23]. The gelsolin treatment was repeated until no thin filaments were visible. The quality and concentration of thick filaments were assessed by negative staining with 2% uranyl acetate. Grids were plunge-frozen in liquid ethane, cooled by liquid nitrogen, via Vitrobot Mark IV (Thermo Fisher Scientific, Waltham, MA, USA).

### 4.2. Data Collection

Data collection was performed in four separate sessions, using two different facilities. The medium-range resolution dataset (tubular filaments) was collected at FSU, using a Titan Krios G1 microscope operating at 300 keV with a K3 camera. The FSU session collected 16,500 movies at a pixel size of 1.12 Å. This dataset did not yield a structure of sufficient resolution for building an atomic model, prompting us to prepare a second relaxed thick filament sample (solid filaments) using the same preparation method. The high-resolution data (solid and tubular filaments) were collected at the National Center for CryoEM Access and Training (NCCAT), where we employed a Titan Krios G4 electron microscope, operated at 300 keV and equipped with a cold FEG, Falcon4i camera and Selectris energy filter. The solid filament dataset was collected at NCCAT in two separate sessions and comprised 39,660 movies with a pixel size of 0.959 Å. Notably, the myosin heads in the solid filaments appeared ordered, leading us to process this dataset independently from the samples, which had disordered heads (Figure 1A,B)

The results from these two datasets motivated us to investigate the potential impact of aging on head organization and the presence of non-myosin proteins. Consequently, we prepared a third sample (tubular) using relaxed thick filaments from 30-day-old flies, which are considered middle-aged. Initial screenings predominantly showed disordered-head filaments (Figure 1A,B), but a portion exhibited heads that were neither distinctly ordered nor disordered. For this third session, also at NCCAT, we collected 27,365 movies maintaining the pixel size of 0.959 Å.

### 4.3. Data Processing

Movie frames were aligned with dose weighting using Motioncor2 [57]. The CryoSPARC v4.2.0 [58] Patch CTF tool was used to estimate the contrast transfer function (CTF). CryoSPARC software (v4.2.0) was used for all subsequent image-processing steps. After aligning frames and correcting the CTF, we conducted manual picking and 2D classification on a subset of 50 micrographs to create a template for the filament tracer tool. Subsequently, we applied the filament tracer to all micrographs, optimizing parameters to maximize filament detection. To further refine our selections, we fine-tuned various thresholds using the Inspect Particle Picks tool. Filament segments (particles) were extracted with a box size of 768 × 768 pixels, corresponding in length to five 145 Å axial repeats. We opted not to remove more false particles through template picking but focused on refining class averages for homogeneity through multiple rounds of 2D classifications. This resulted in 255,577 segments for Specimen 1, 444,666 segments for Specimen 2, and 522,004 segments for Specimen 3, which we used for subsequent refinement.

The image-processing approach for both disordered- and ordered-head datasets was almost identical. The structure was reconstructed under C4 rotational symmetry using a reconstruction of the *Lethocerous* flight muscle thick filament as a reference, low-pass filtered to 60 Å (*Lethocerus* has the same axial rise as *Drosophila*). Multiple rounds of homogenous refinement coupled with local CTF refinement were performed on the selected segments. To further improve the resolution and increase the quality of the map, non-uniform refinement (NU-refine) was performed. In addition to an improvement in the FSC, the map quality was visibly enhanced, providing a clearer and more detailed structure. Segmentation was performed in Chimera [59].

## 5. Conclusions

Insect indirect flight muscles (IFMs) attach to the cuticle of the thorax instead of directly to the wings. Flight is achieved via IFM contractions at the thorax resonant frequency. Insects utilizing IFM diverged from a common ancestor 370 Mya into four orders: true bugs (Hemiptera), flies (Diptera), bees (Hymenoptera), and beetles (Coleoptera). Muscles conserve ATP by myosin heads assuming an ordered arrangement, a super-relaxed state. Two myosin head arrangements, ordered and disordered, characterize IFM myosin filaments. Disordered forms are found in the Asian bumble bee and fruit fly. The giant water bug, which lives mostly underwater, produces the ordered form. Here, we show that the fruit fly can form a similar ordered-head arrangement, possibly to compensate for mitochondria degradation with age. Vertebrate skeletal muscle alters protein expression when changing from one fiber type to another; here, we have observed a similar change in patterns of protein expression in changing from the ordered-head to the disordered-head filament form in *Drosophila* flight muscle. There seems to be no obvious “switch” in the other muscle proteins common to both thick filament forms that leads to the change in filament structure.

## Figures and Tables

**Figure 1 ijms-25-11313-f001:**
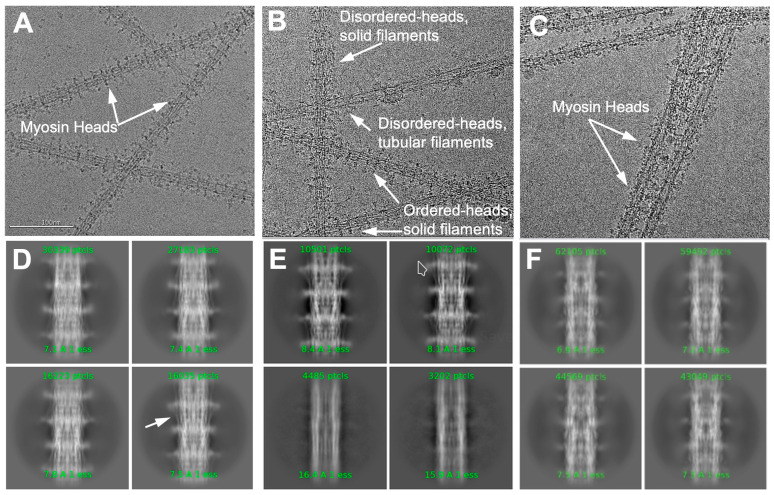
Original micrographs and class averages illustrating the two filament structures. (**A**) Solid filaments showing ordered myosin heads. (**B**) Filaments showing heterogeneous head arrangements including solid filaments with ordered and disordered heads and tubular filaments with disordered heads. (**C**) Tubular filaments with disordered heads from 30-day-old flies. (**D**) Class averages from solid filaments showing enhanced details of heads and solid backbones. White arrow points to a visible segment of S2, the initial segment of the myosin tail. (**E**) Additional class averages from the solid filament dataset showing the enhanced features of ordered heads with solid backbones (top) and disordered heads with hollow backbones. The bottom pair of class averages show almost no myosin head density and a hollow core. (**F**) Class averages from the tubular dataset of 30-day-old flies showing weak head density and a hollow filament core.

**Figure 2 ijms-25-11313-f002:**
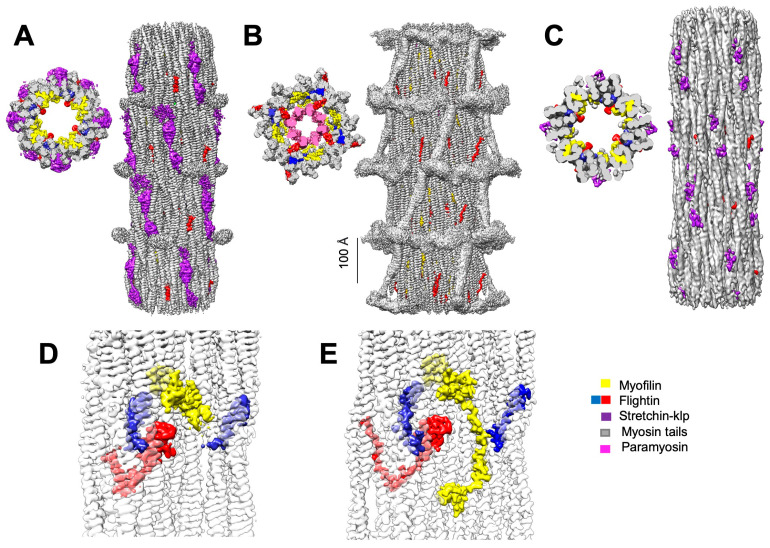
(**A**–**C**) Detailed longitudinal and cross-sectional views of segmented reconstructions showing non-myosin protein distribution in disordered- (**A**) and ordered-head (**B**) filaments, with a notable presence or absence of stretchin-klp on the surface and paramyosin in the core. (**C**) The 7.88 Å tubular filament segmented map generated from a class of filaments from the solid filament dataset showing stretchin-klp on the backbone surface and the absence of paramyosin in the core. (**D**,**E**) Exploration of the distribution of non-myosin proteins among myosin tails indicating structural variations between disordered- (**D**) and ordered-head (**E**) filaments. Density attributable to myofilin is compact when the heads are disordered (**D**) but larger and more extended when the myosin heads are ordered (**E**). Note also the greater size and difference in shape of the flightin C-terminal domain density (blue) in the solid filaments (**E**) and the more compact shape in the tubular filaments (**D**).

**Figure 3 ijms-25-11313-f003:**
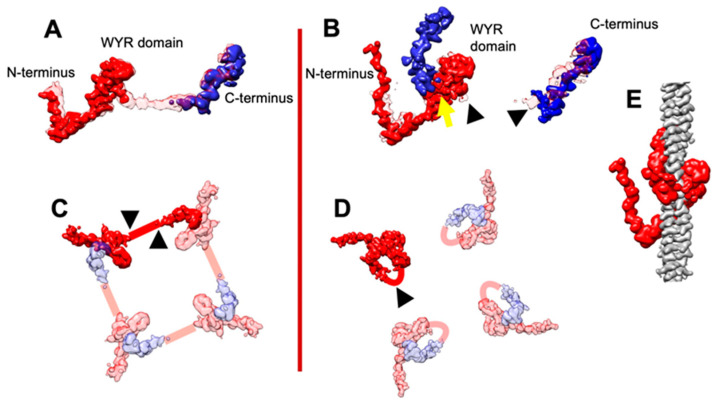
Flightin structural variations. (**A**,**B**) Flightin density comparisons. (**A**) shows the tubular filament flightin (solid) superimposed onto the flightin from the 4.7 Å map (transparent), where the blue and red densities are connected. A trace of a possible connection is present in the tubular filament flightin reconstruction from the present work, but at no contour threshold can a connection be seen. (**B**) The solid filament flightin superimposed on the tubular filament flightin (transparent). The tubular filament’s possible connection (black arrowheads) is missing in the solid filament version, but a new potential link (yellow arrowhead) is suggested. (**C**,**D**) Illustration of flightin’s shape differences. (**C**) In the tubular filaments, the red density containing the WYR domain and blue density containing the C-terminus are distant from each other, giving their connection, defined by the black arrowheads, an elongated shape. (**D**) In the solid filaments, the red and blue densities are juxtaposed, with the connection (black arrowhead) close to the WYR domain creating a compact flightin shape. (**E**) A low-pass-filtered density of the compact flightin structure, in which the WYR and C-terminal domains, both colored red, literally wrap around a myosin tail. In the extended flightin structure, these domains are in the same location but come from separate flightin molecules.

**Figure 4 ijms-25-11313-f004:**
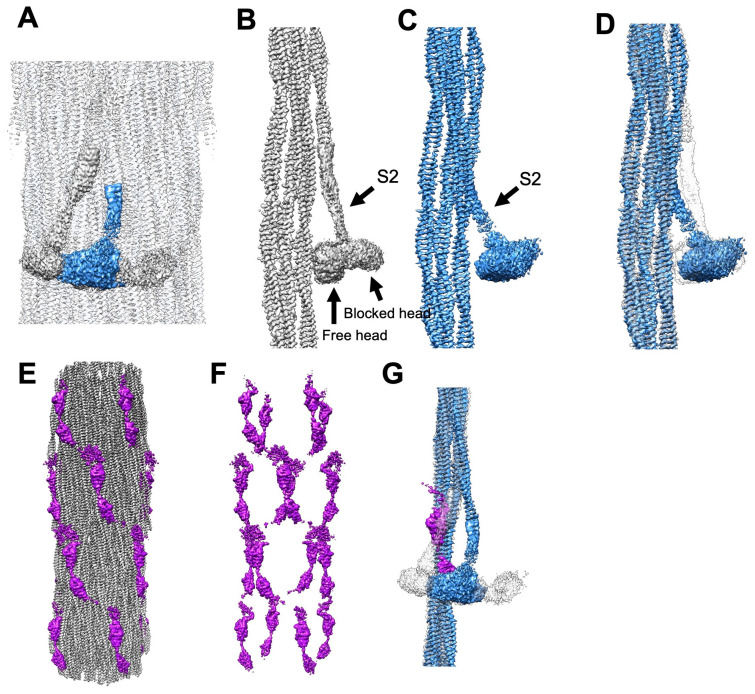
Stretchin-klp, myosin heads, and S2 densities in solid and tubular filaments. (**A**) Longitudinal view of the solid filament reconstruction (gray) with superposition of myosin head and proximal S2 densities from the tubular filaments (blue). (**B**,**C**) Myosin tail layer with proximal S2 average head density from solid filaments (gray) and tubular filaments (blue). (**D**) Views B and C are superimposed showing the average head and S2 density of tubular filaments (blue) and solid filaments (gray, transparent). (**E**–**G**) Stretchin-klp from tubular filaments. (**E**,**F**) Stretchin-klp (purple) on the outside of the *Drosophila* tubular filament following a left-handed helical track across the right-handed myosin helical symmetry. (**G**) Demonstration of stretchin-klp (purple) helical tracks passing under the location where the IHM “free” head binds the filament backbone, indicating how stretchin-klp potentially blocks free-head binding in tubular filaments.

**Figure 5 ijms-25-11313-f005:**
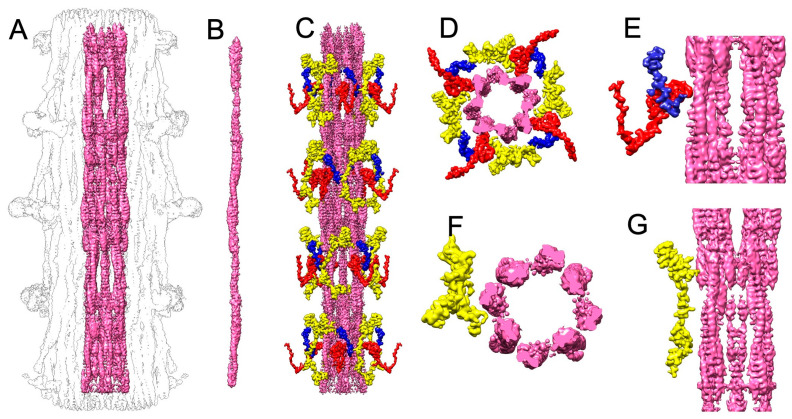
Paramyosin. (**A**) Paramyosin in the solid filament’s core, featuring thicker coiled coils near the crown level and a thinner region between the crowns. Color scheme: paramyosin, pink; flightin, red (WYR and adjacent domains) and blue (C-terminus); myofilin, yellow. (**B**) Segmented, single paramyosin coiled coil. (**C**,**D**) Side view and top view, respectively, of paramyosin interactions with non-myosin densities in the filament’s core. (**E**) Flightin densities go into the paramyosin core, seemingly contacting its outer part, yet higher magnification shows no actual connection to the paramyosin. (**F**,**G**) Top and side views show that myofilin density does not contact paramyosin at this resolution.

## Data Availability

Full reconstructions of the *Drosophila melanogaster* thick filament are available in the EMDB under accession codes EMD-47369 for the disordered-head structure and EMD-47368 for the ordered-head structure.

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
