# Peer review of "Two Forms of Thick Filament in the Flight Muscle of Drosophila melanogaster"

_ijms, 2024, doi:10.3390/ijms252011313_

Round 1
Reviewer 1 Report
Comments and Suggestions for Authors
In this manuscript titled “Two forms of thick filament in the flight muscle of Drosophila melanogaster” by Rastegarpouyani, Hojjatian, and Taylor, the authors present a new cryo-EM analysis of thick filament structure in the fruit fly. The basis of the study is the observation of two distinct populations of thick filaments in the data: those with larger hollow cores referred to as tubular filaments and those that are less-hollow referred to as solid filaments. Using their cryo-EM data, the authors show that tubular filaments have disordered myosin heads which corresponds to the presence of Stretchin-klp. By contrast, solid filaments with a paramyosin core (a complete internal annulus of Prm) have ordered myosin heads and lack high levels of Stretchin-klp. In addition, the authors describe two different conformations of Flightin, with an extended conformation found in tubular filaments and a compact conformation found in solid filaments. A fourth protein, myofilin, also changes conformation, having a compact conformation when heads are disordered and a more extended conformation when the heads are ordered.
This manuscript extends previous models of thick filaments determined at 4 to 7Å to a resolution of 2.8Å. There are several densities, interactions, and domains where this increased resolution offers advantages over previous structures. This manuscript also offers insight into the “tubular” and “solid” thick filament types that are reported in previous EM and structural analyses from various insects. The use of Drosophila is appropriate as a model. The manuscript reports an advance in the field that will be of interest to both structural and muscle biologists. I have the following suggestions to improve the manuscript:
Line 49 – Can the authors add a sentence clarifying the difference between Stretchin-klp and Stretchin-Mlck (Strn-Mlck, CG44162)? Given the various splice isoforms of Stretchin, are any isoforms sensitive to calpain digestion (similar to how kettin is sensitive), that they could be associated with the solid filaments but lost during preparation?
Are there any data from Drosophila to suggest if “tubular” and solid filaments are found in the same sarcomere? Genetic labeling and microscopy approaches do not reveal any gaps in the Stretchin-Mlck staining pattern in IFMs, but it is possible that mixed filaments are beyond the limits of light microscopy.
Line 52 is confusing. Thick filaments are anchored to the M-line, while thin filaments are anchored at the Z-disc. The connecting filament made of projectin/kettin do link the thick and thin filaments, and connecting filaments are anchored at the Z-disc, but the sentence as written makes it seem thick filaments are anchored to the Z-disc.
In line 85-86, it is not clear what “except to exclude the generally smaller male flies” means. This is also not obvious from the methods. I appreciate that the age of the sampled flies is unknown, but were they male or female or mixed flies? If males were removed, are these thick filaments from mixed-age females?
In line 252, how do the data support the Fln, Mf, and Prm are important in determining filament length? Otherwise, please cite the relevant genetic study.
In lines 325-331, the authors discuss the relationship between age and tubular vs solid thick filament structure. This should be presented clearly as a hypothesis, given the lack of clarity on the age of the sample. Would a newly eclosed fly also have solid thick filaments?
In line 347 and again in 363-364, it isn’t clear to me how a potential transition to a complete internal annulus of Prm (ie tubular to solid thick filaments) would lead to a stiffer muscle fiber. How would this result in the observation in the X-ray diffraction that myosin heads moved towards the thin filaments? How does the data here and a potentially stiffer thick filament affect muscle stiffness and sarcomere contractility?
Is there any other types of data that might support the transition to solid thick filaments (or that proportionally more thick filaments are solid) in older flies? For example, is there more expression of Prm or more Prm incorporated into myofibrils/sarcomeres in older flies? Does the difference in Prm structure actually reflect different isoforms of Prm? It is known that Mf, Prm, and Mhc itself have different isoforms which are developmentally and fiber-type specifically regulated. How would these types of changes in the isoform of these key structural proteins affect the cryo-EM structures a different points in time? Do the expression data support the hypothesis proposed here?
In the Materials and Methods, please provide more details in the specimen preparation section. The dissection protocol needs to be described, and the age and sex of the flies should be reported. Were the IFMs dissected in buffer (relaxing buffer?)? Were flies dissected with needles, forceps, scissors, or another method? Were female or male IFM analyzed?
I understand the context in the field of naming tubular versus solid thick filaments. I just want to note that this can be confusing, considering that body muscles are also referred to as tubular muscles (while the IFM are fibrillar). Is it known what the structure of the thick filaments are in tubular muscles? Are there “solid” or “tubular” thick filaments in tubular muscles?
Legend for Figure 5 is not visible (cut off by text box).
Reviewer 2 Report
Comments and Suggestions for Authors
This work study the insect indirect flight muscle (IFM), where the flight powering
muscles are not directly attached to the wings. The authors used cryo-electron microscopy of isolated IFM myosin filaments from Drosophila melanogaster. Based on large size of imaging data they found two distinct filament types: one with a tubular backbone and another with a solid backbone. Tubular filaments lacked visible densities within the annulus formed by myosin tails and solid filaments had four paired paramyosin densities. The myosin heads in tubular filaments were disordered, whereas solid filaments had one head fully immobilized and another partially immobilized. Tubular filaments were coated with stretchin-klp, which solid filaments lacked. Flightin and myofilin were present in all filaments, with flightin adopting compact and extended conformations in solid and tubular filaments, respectively.
Solid filaments in Lethocerus indicus suggest that both species may conserve ATP by using the super-relaxed state.
The paper shows an original contribution in the area of Drosophila flight muscle and provides new insights into the structural properties of IFM by using a technical advanced method with high resolution (CryoEM). The subject matter falls well within the scope of "IJMS" journal and the Special Issue. The methodology is appropriate and the text flow is well organized. Therefore, I recommend the current manuscript for further publication.
Hereafter, some minor comments for authors to consider in their revision.
1. The statement "Nearly identical solid filaments occur in the large water bug Lethocerus indicus, …., suggesting solid filaments conserve ATP utilizing the super relaxed state" is speculative and a bit too strong to be in the abstract.
2. The text does not clearly specify if the analysis is based on a mix of male or female flies of similar body size (line 86). The authors should clarify this. Additionally, if the two forms of thick filaments in flight muscle are structurally similar between males and females, it should be explicitly stated. If any sexual dimorphism in filament structure was observed, it should be highlighted in the text.
3. The growing conditions, including specifics about the diet (food recipe), are not provided. Effects of fasting and subsequent feeding on indirect flight muscle (IFM) alterations have been earlier reported. Considering the significant influence of diet on energy metabolism and aging, these details are important and should be included. The authors must ensure that the growing conditions are comparable to other studies to validate the findings related to aging.
4. The observation that adult Drosophila thick filaments transition from exclusively tubular to predominantly solid is intriguing but not systematically examined in this study. More age points are needed.
5. The use of IFM from 30-day-old flies should be indicated in the figure legends (Figure 1) to make it easier for readers to follow and understand the data presented.
Reviewer 3 Report
Comments and Suggestions for Authors
Review attached
